# Outcomes of Deep Sclerectomy for Glaucoma Secondary to Sturge–Weber Syndrome

**DOI:** 10.3390/jcm12020516

**Published:** 2023-01-08

**Authors:** Faisal A. Almobarak, Abdullah S. Alobaidan, Mansour A. Alobrah

**Affiliations:** 1Department of Ophthalmology, College of Medicine, King Saud University, Riyadh 11411, Saudi Arabia; 2Glaucoma Research Chair, King Saud University, Riyadh 11411, Saudi Arabia

**Keywords:** glaucoma, Sturge–Weber syndrome, deep sclerectomy, intraocular pressure, filtering surgery, glaucoma surgery

## Abstract

Aims: To report the outcomes and complications of deep sclerectomy in glaucoma secondary to Sturge–Weber syndrome (SWS). Methods: The retrospective case series included patients with SWS and secondary glaucoma who underwent deep sclerectomy at King Abdul Aziz University Hospital, Riyadh, Saudi Arabia between 2000 and 2021. The main outcome measures included intraocular pressure (IOP), the number of antiglaucoma medications, the presence of vision-threatening complications, and the need for further glaucoma surgery to control the IOP. The surgical outcome of each eye was based on the main outcome measures. Results: Twelve eyes of eleven patients were included in the study. The mean follow-up period was 83.00 months (±74.2) (range 1 to 251 months). The IOP and number of antiglaucoma medications decreased significantly from a mean of 28.75 mm Hg (±7.4) and 3.17 (±0.8) to 15.30 mm Hg (±3.5) and 0.3 (±0.7), and 18.83 (±9.3) and 1.67 (±1.7) on the 24th month and the last follow-up visit postoperatively, respectively (*p* < 0.01 for both). The success rate was 66.6% (8/12), while the failure rate was 33.3% (4/12) because of the uncontrolled IOP where a single repeat glaucoma surgery achieved controlled IOP. One procedure was complicated by choroidal detachment and one by choroidal effusion; both complications were resolved by medical treatments. Conclusions: Deep sclerectomy seems to be an effective treatment modality for controlling IOP and for decreasing the burden of antiglaucoma medications in patients with SWS and secondary glaucoma. Further studies are needed to confirm such a conclusion on larger number of patients with longer follow-up periods.

## 1. Introduction

Sturge–Weber syndrome (SWS) is a rare, sporadic, congenital neurocutaneous disorder that affects the brain, skin, and eyes. It is considered one of the phacomatoses that develop because of neural crest anomalies, such as neurofibromatosis, Klippel–Trenaunay syndrome, tuberous sclerosis, and von Hippel–Lindau syndrome. SWS is characterized by leptomeningeal hemangioma, facial angiomatosis, or nevus flammeus (port-wine stain) in the ophthalmic division of the trigeminal nerve, and ocular changes, such as glaucoma and choroidal hemangioma. The estimated incidence is between 1:20,000 and 1:50,000 infants, with no significant difference between the sexes [1]. Recently, somatic mosaic mutations in the *GNAQ* gene located on the long arm of chromosome 9 have been reported as a part of the genetic basis of SWS resulting in capillary malformations [2].

The incidence of glaucoma in patients with SWS varies between 30% and 70% [3]. The presentation and pathogenesis of glaucoma can be divided into two main categories: childhood and adulthood presentation. The early childhood or infancy presentation mainly develops because of outflow obstructions associated with angle malformations such as those observed in congenital glaucoma, while the adulthood presentation develops because of elevated episcleral venous pressure arising from vascular malformations that impede outflow and accelerated aging of the angle structures [4,5]. The conventional aqueous outflow faces the highest resistance in the juxtacanalicular trabecular meshwork. However, in SWS, the dilated episcleral vessels show a further, abnormal pressure gradient in the conventional pathway caused by vascular malformation [6]. Medical management seems to be less effective in SWS, and it cannot guarantee a good long-term control of glaucoma, especially for the childhood type [7,8]. When medical treatment fails to halt glaucoma progression, surgical intervention is needed. However, owing to the rare nature of the disease, few studies are available on the outcome of different surgical modalities; therefore, there is inconsistency in such interventions. Generally, the decision is based on the angle’s status: with respect to whether it is open or closed. Trabeculectomy with or without trabeculotomy has been proposed as a choice for creating a new outflow tract for the aqueous humor, but it carries a significant risk of vision-threatening complications in patients with SWS which include massive choroidal detachment and expulsive hemorrhage [9,10]. Deep sclerectomy is a non-penetrating glaucoma surgery that allows a gradual egress of aqueous through the trabeculo-descemet’s window (TDW) and, therefore, avoids hypotony and other choroidal-related complications in trabeculectomy. The efficacy and safety of deep sclerectomy have been previously described, even in patients with congenital glaucoma [11]. Nevertheless, there is insufficient evidence regarding the efficacy of deep sclerectomy in treating SWS. To our knowledge, only one study has reported the outcome of deep sclerectomy in patients with SWS [12]. Therefore, this study aimed to report the outcomes of deep sclerectomy in eyes with glaucoma secondary to SWS.

## 2. Materials and Methods

### 2.1. Patients

We reviewed the medical records of patients with SWS and secondary glaucoma who underwent deep sclerectomy at King Abdul Aziz University Hospital, Riyadh, Saudi Arabia. The study was approved by the Institutional Review Board (E-20-4996) of King Saud University as a part of a larger study on the outcomes of deep sclerectomy, and all procedures adhered to the tenets of the Declaration of Helsinki. In early childhood glaucoma, examinations under general anesthesia were performed, and the following observations were documented: the intraocular pressure (IOP) (measured using the Perkins applanation tonometer), horizontal corneal diameter, corneal edema, central corneal thickness, and cup-to-disc ratio, and dilated fundus examination. Surgery was performed when the patient was confirmed to have glaucoma. In patients with later presentation adulthood glaucoma, surgery was performed when the patient had the following: (i) medically uncontrolled IOP of ≥21 mmHg (measured using the Goldmann applanation tonometer) despite the maximum number of tolerated antiglaucoma medications and (ii) the presence of progressive glaucomatous optic nerve head damage.

### 2.2. Surgical Methods

One of several staff glaucoma specialists credentialed for the procedure performed the surgeries in accordance with standard documented techniques under general anesthesia. First, a fornix-based conjunctival peritomy was performed, and hemostasis of the episclera was achieved. A 4 × 5 mm superficial scleral flap was dissected toward the cornea, and 0.2 mg/mL of mitomycin C (MMC) soaked in a sponge was applied under the flap and conjunctiva for 2 min followed by balanced salt solution irrigation. A deeper scleral flap was dissected approximately 0.5 mm internal to the edges of the superficial flap right above the choroid, and a TDW was created, followed by deroofing of Schlemm’s canal. When micro-perforations occurred in the TDW but with a normal depth anterior chamber, the procedure was completed as planned. When the TDW was perforated along with iris prolapse, peripheral iridectomy was performed, and the surgery was converted to a trabeculectomy. Thereafter, the deep flap was excised, and the superficial flap was closed using two 10.0 monofilament nylon sutures. The conjunctiva was closed using 9.0 vicryl sutures, and the wound was checked for the presence of leakage. Finally, antibiotics and steroids were subconjunctivally injected. After surgery, all patients were treated with topical ofloxacin and prednisolone acetate 1% drops.

### 2.3. Data Analysis

Postoperative visits were defined as those made on the first postoperative day; at 2–4 weeks, 3 months, 6 months, and 12 months postoperatively; and yearly thereafter. Pre and postoperative data were collected whenever available and applicable for the following variables: age at diagnosis and the time of surgery, sex, IOP, number of antiglaucoma medications, best-corrected visual acuity converted into logarithm of minimal angle of resolution format when available, time to failure, postoperative complications, and need for subsequent pressure-lowering procedures to control the IOP.

The variables were evaluated using Student’s *t*-test and the Wilcoxon rank test and presented as means and standard deviations (SD). *p* values of <0.05 were considered significant. Surgical success was classified as follows: (i) complete success (IOP reduction of ≥20% from the baseline level or IOP between 6 and 21 mmHg without antiglaucoma medications, no loss of vision due to glaucoma progression, no postoperative vision-threatening complications, and no need for further glaucoma procedures to control the IOP); (ii) qualified success (IOP reduction of ≥20% from the baseline level or IOP between 6 and 21 mmHg with antiglaucoma medications, no loss of vision due to glaucoma progression, no postoperative vision-threatening complications, and no need for further glaucoma procedures to control the IOP); and (iii) failure (IOP reduction of <20% from the baseline level or IOP of >21 mmHg despite a maximum number of tolerated antiglaucoma medications on two visits, persistent hypotony [IOP of ≤5 mmHg] on two visits causing hypotony maculopathy, loss of vision due to glaucoma progression, postoperative vision-threatening complications, or the need for further glaucoma procedures to control the IOP). The cumulative probabilities of overall success, presented as percentages ± standard errors, were determined via a Kaplan–Meier life table analyses. Statistical analyses were carried out using SPSS version 23 (SPSS Inc., Chicago, IL, USA).

## 3. Results

### 3.1. Patient Characteristics

Eleven patients, including six boys and five girls (12 eyes), underwent deep sclerectomy as the first procedure for glaucoma secondary to SWS. One patient underwent a bilateral surgery, while the remaining patients underwent a unilateral surgery. The mean age at the time of surgery was 5.58 (±68.3) years (range, 2 months to 14 years). Three eyes (25%) had associated choroidal hemangioma. In all patients, there was no history of any ophthalmic surgery other than deep sclerectomy. The mean follow-up time was 83.00 (±74.2) months (range, 1–251 months). Most eyes had advanced glaucomatous disc damage (75%) (Table 1).

### 3.2. Efficacy

The IOP decreased from a preoperative baseline of 28.75 (±7.4) mmHg to 14.90 (±3.9), 15.30 (±3.5), 15.57 (±4.3), 20.00 (±7.6), and 18.83 (±9.3) mmHg, while the number of antiglaucoma medications decreased from a preoperative baseline of 3.17 (±0.8) to 0.13 (±0.4), 0.30 (±0.7), 0.71 (±1.3), 1.00 (±1.3), and 1.67 (±1.7) on the 12th month, 24th month, 36th month, 48th month, and the last follow-up visit postoperatively, respectively (all *p* < 0.05). The cumulative probabilities of overall success were 75.0% (±12.5%) on the 12th month and 24th month, 66.7% (±13.6%) on the 36th month and 48th month, and 58.3% (±14.2%) on the 60th month postoperatively (Figure 1). Over the entire follow-up period, controlled IOP was achieved in eight eyes (66.6%) after deep sclerectomy: complete success in five eyes (41.7%) and qualified success in three eyes (25.0%). Meanwhile, the treatment failed in four eyes (33.3%) because of uncontrolled IOP, despite the maximum number of tolerated antiglaucoma medications and required additional glaucoma surgery to control the IOP: Ahmed implantation in two eyes and deep sclerectomy in two eyes. The mean failure time was 75.0 (±78.9) months (range, 2–187 months). One eye failed within the first 3 months after deep sclerectomy, while the remaining eyes failed after 50 months. All four eyes achieved IOP control after a single repeat glaucoma surgery.

### 3.3. Safety

Intraoperative complications occurred in four eyes: two eyes had TDW perforations, but without iris prolapse, wherein the procedure was continued while maintaining a formed anterior chamber; two eyes had TDW perforation with iris prolapse, wherein peripheral iridectomy and iris repositioning were performed without sclerotomy. The deep flap was excised, and the procedure was continued. Postoperative choroidal effusions occurred in one eye with existing choroidal hemangioma, while isolated choroidal detachments occurred in one eye with no choroidal hemangioma. Both conditions resolved within 1 month after medical treatment including topical steroids and cycloplegic agents.

## 4. Discussion

Glaucoma that is secondary to SWS is a complex disease. The etiology has been attributed to trabecular meshwork anomalies for early-onset glaucoma and elevated episcleral venous pressure due to arteriovenous shunts and the premature aging of the trabeculae for juvenile/adult-onset glaucoma [13,14]. Medical therapy usually fails to achieve long-term IOP control; therefore, surgery is required to avoid visual loss. The type of surgical intervention usually depends on the pathophysiology of the disease. In SWS, deep sclerectomy is a preferred option, as it creates a new outflow tract, therefore avoiding episcleral venous pressure elevation. Furthermore, deep sclerectomy has been reported to be successful in treating primary congenital glaucoma (PCG). Eyes with PCG and glaucoma secondary to SWS have been shown to have trabecular meshwork anomalies; however, such anomalies seem less severe in patients with SWS than in those with PCG [15]. To the best of our knowledge, only one study has evaluated the outcomes of deep sclerectomy for glaucoma secondary to SWS [12]. The current study showed that deep sclerectomy is effective as an initial procedure for glaucoma secondary to SWS.

Herein, the IOP and number of antiglaucoma medications decreased significantly during the first year and the years thereafter. Complete success was achieved in 41.7% of the eyes and qualified success in 25.0% of the eyes on the last follow-up visit. Audren et al. reported a complete success rate of 56%, 28%, and 0% at 6, 13, and 26 months after deep sclerectomy, respectively. Our study showed a comparable survival rate in the short-term among the eyes that received Ahmed implants, but the results show better long-term survival rates. In their study that included 11 eyes, Hamush et al. reported a 2-year success rate of 79% and a 5-year success rate of 30% after using Ahmed implants. Kaushik et al. reported a 24-month success rate of 75% in 24 eyes that received Ahmed implants with a mean follow-up period of 2.12 years [16]. However, long-term survival seems to be better in the current study compared with both studies. Herein, the IOP increased over the follow-up period, and some eyes needed more antiglaucoma medications over time to control the IOP; this case is well known in bleb-dependent filtering surgery, which carries a high risk for surgical failure. Furthermore, the effect of a higher concentration of MMC is not expected to contribute to the surgical success and IOP control [17,18]. Other studies have reported encouraging outcomes of trabeculotomy and goniotomy. Olsen et al. [19] reported a success rate of 66.7% in 14 patients with early-onset glaucoma secondary to SWS after a mean follow-up of 5.4 years after one or more procedures, while Wu et al. [15]. reported a 1-year success rate of 86.6% in 32 patients; both results are comparable to our report. Of the four failed eyes herein, one eye with juvenile-onset glaucoma failed within the first 3 months and achieved IOP control after Ahmed implants; meanwhile, three eyes with early-onset glaucoma failed after 50 months and achieved IOP control after repeat deep sclerectomy (two eyes) and Ahmed implant (one eye).

Postoperative choroidal detachment occurred in one eye with intraoperative TDW perforation; choroidal effusion occurred in one eye with intraoperative TDW perforation and iris prolapse and co-existing choroidal hemangioma, wherein iridectomy was performed. No expulsive choroidal hemorrhage occurred. Both complications were self-limiting and responded well to medical treatments in the form of topical atropine and prednisolone drops. In SWS, the presence of aberrant clusters of capillary–venule-like blood vessels, choroidal vessel’s overgrowth and thickening, and overabundant vessels increases not only the risk of glaucoma but also that of choroidal effusion. Since the mitogen-activated kinase pathway does not show significantly increased signaling activities in cells with *GNAQ*, the β-blocker propranolol could not have a crucial role in such cases [20]. Nevertheless, in intractable choroidal effusion failing to respond to medical treatments, propranolol could still have a long-term role in resolving such complications [21]. Although bleb-dependent procedures for SWS are associated with high rates of complications, such as expulsive hemorrhage and prolonged hypotony, none of these complications were observed in the current study [6]. However, a major advantage of deep sclerectomy is the higher safety profile and fewer postoperative complications because of the progressive outflow of aqueous through an intact TDW, which precludes complications, such as sudden hypotony, aqueous misdirection, and expulsive hemorrhage [22]. Adding a control group that underwent other surgical modalities in the current study to compare the IOP control and complications would be ideal. However, this was not possible because of the rare nature of the disease.

## 5. Conclusions

Although the current retrospective study included a small number of eyes because of the rare nature of the disease, it showed the efficacy of deep sclerectomy in treating glaucoma secondary to SWS. All complications were self-limiting and responded to medical treatments. Further studies are needed to compare the outcomes of deep sclerectomy with those of other filtering procedures.

## Figures and Tables

**Figure 1 jcm-12-00516-f001:**
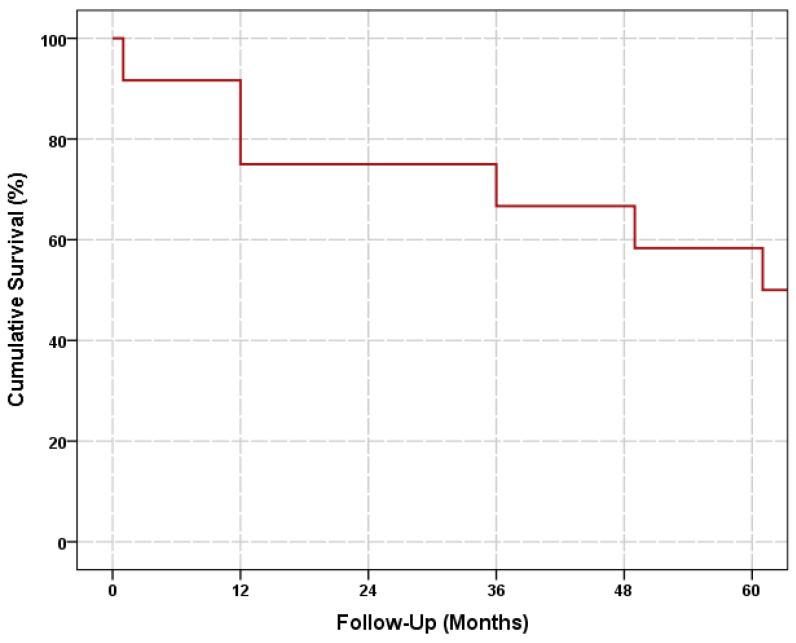
Kaplan–Meier survival curves showing the cumulative probability of success.

**Table 1 jcm-12-00516-t001:** Patient characteristics, complications, and success.

Patient	Eye	Age at Surgery	Preoperative IOP	No. of Preoperative Medications	Intraoperative Complications	Postoperative Complications	Follow-Up Duration	Status	Final IOP	No. of Final Medications
1	OD	2 months	43	4			7.8 years	Complete success	13	0
	OS	2 months	39	4			7.8 years	Complete success	13	0
2	OS	10 months	30	2	TDW perforation with iris prolapse		15.6 years	Failed	22	3
3	OD	8.5 years	28	4			1 month	Failed	37	2
4	OS	9.5 years	33	4	TDW perforation with iris prolapse	Choroidal detachment	7.8 years	Qualified success	16	4
5	OD	3 months	32	2	TDW perforation	Choroidal effusion	5.1 years	Failed	30	3
6	OS	1.3 years	25	3			4.1 years	Failed	32	3
7	OS	2 years	26	2	TDW perforation with iris prolapse		20.9 years	Qualified success	14	4
8	OD	8.3 years	20	3			8.8 years	Qualified success	18	1
9	OD	12.4 years	30	3			3.7 years	Complete success	12	0
10	OD	13.3 years	20	3			1 year	Complete success	10	0
11	OS	14.2 years	19	4			1 year	Complete success	9	0

OS: left eye; OD: right eye; IOP: intraocular pressure; TDW: trabeculo-descemet window.

## Data Availability

All data are available from the corresponding author upon request.

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
