# Peer review of "Outcomes of Deep Sclerectomy for Glaucoma Secondary to Sturge–Weber Syndrome"

_jcm, 2023, doi:10.3390/jcm12020516_

Round 1
Reviewer 1 Report
The authors evaluated the efficacy and safety of DS in treatment of glaucoma secondary to SWS. The manuscript is interesting and can help us better understand the treatment options of SWS glaucoma. However,
1. For this is a retrospective study, I suggest to add a control group of other treatment.
2. I suggest the authors briefly discuss the outcomes of other treatment in addition to Ahmed glaucoma valve implantation (trabeculectomy, goniosurgery, etc) in section ‘Discussion’.
3. There are too many typo or grammatical errors, please proofread the manuscript carefully.
Author Response
- For this is a retrospective study, I suggest to add a control group of other treatment.
Response: We thank the reviewer for bringing up this point. This will actually strengthen our study more. However, because of the rare nature of the disease, adding a control group of SWS patients who underwent other treatments was not possible. Our existing data is very limited because of such a rare nature.
- I suggest the authors briefly discuss the outcomes of other treatment in addition to Ahmed glaucoma valve implantation (trabeculectomy, goniosurgery, etc) in section ‘Discussion’.
Response: Done and added.
- There are too many typo or grammatical errors, please proofread the manuscript carefully.
Response: Proofreading was done using MDPI services and external services as well, and all errors were corrected.

Reviewer 2 Report
Original article with some degree of interest
line 86: Introduction. It is good to complete the introduction with a brief description of the adult's form as well
line 103: optic nerve head damage or progressive damage?
Line 139: Because of the young age of the patients, the method of measuring IOP should be specified
tab: patient #10 and 11: follow-up is too short for us to claim "complete success"
Line 257: Long-term efficacy ONLY for some patients, for others this claim cannot be made
Author Response
- line 86: Introduction. It is good to complete the introduction with a brief description of the adult's form as well
Response: Done and added. The paragraph now reads:
The early childhood or infancy presentation mainly develop because of outflow obstruction associated with angle malformation like in congenital glaucoma, while the later in life adulthood presentation develop because of elevated episcleral venous pressure arising from vascular malformations that impede outflow, and the accelerated aging of the angle structures. The conventional aqueous outflow faces the highest resistance in the juxtacanalicular trabecular meshwork. But in SWS, the dilated episcleral vessels indicate a further, abnormal pressure gradient in the conventional pathway caused by the vascular malformation.
- line 103: optic nerve head damage or progressive damage?
Response: We thank the reviewer for raising this point. Error was corrected. The sentence now reads:
(ii) the presence of progressive glaucomatous optic nerve head damage.
- Line 139: Because of the young age of the patients, the method of measuring IOP should be specified
Response: The methods for measuring the IOP were Perkins applanation tonometer for early onset glaucoma, and Goldman applanation tonometer for patients with late presentation glaucoma. Both were added to the methods section.
- tab: patient #10 and 11: follow-up is too short for us to claim "complete success"
Response: We thank the reviewer for bringing up this point. This was a copy error which was corrected. The follow-up time for both was 1 year as can be even concluded from the survival curve.
- Line 257: Long-term efficacy ONLY for some patients, for others this claim cannot be made
Response: We thank the reviewer for bringing up this point. We do agree that such conclusion should not be generalized as the follow-up is short for some patients. We deleted (long-term) and kept it as efficacy alone. The sentence now reads:
Although the current study represents a retrospective study including a small number of eyes because of the rare nature of the disease, it showed the efficacy of deep sclerectomy in glaucoma secondary to SWS.
We modified the conclusion section in the abstract as well which now reads:
Conclusions: Deep sclerectomy seems to be an effective treatment modality to control the IOP and decrease the burden of antiglaucoma medications in patients with SWS and secondary glaucoma. Further studies are needed to confirm such a conclusion on larger number of patients with longer follow-ups.
